# Effects of Combined Allogenic Adipose Stem Cells and Hyperbaric Oxygenation Treatment on Pathogenesis of Osteoarthritis in Knee Joint Induced by Monoiodoacetate

**DOI:** 10.3390/ijms23147695

**Published:** 2022-07-12

**Authors:** Aleksandar Juskovic, Marina Nikolic, Biljana Ljujic, Aleksandar Matic, Vladimir Zivkovic, Ksenija Vucicevic, Zoran Milosavljevic, Radisa Vojinovic, Nemanja Jovicic, Suzana Zivanovic, Nevena Milivojevic, Vladimir Jakovljevic, Sergey Bolevich, Marina Miletic Kovacevic

**Affiliations:** 1Department of Orthopaedic Surgery, Clinical Centre of Montenegro, 81110 Podgorica, Montenegro; aleksandar.juskovic@gmail.com; 2Department of Physiology, Faculty of Medical Sciences, University of Kragujevac, 34000 Kragujevac, Serbia; marina.rankovic.95@gmail.com (M.N.); vladimirziv@gmail.com (V.Z.); drvladakgbg@yahoo.com (V.J.); 3Department of Genetics, Faculty of Medical Sciences, University of Kragujevac, 34000 Kragujevac, Serbia; 4Department of Surgery, Faculty of Medical Sciences, University of Kragujevac, 34000 Kragujevac, Serbia; maticaleksandar@gmail.com; 5University Clinical Center, 34000 Kragujevac, Serbia; rhvojinovic@gmail.com; 6Department of Pharmacology of the Institute of Biodesign and Complex System Modelling, First Moscow State Medical University I.M. Sechenov, 119991 Moscow, Russia; 7Department of Pharmacy, Faculty of Medical Sciences, University of Kragujevac, 34000 Kragujevac, Serbia; ksenija.vucicevic.kg@gmail.com; 8Department of Histology and Embriology, Faculty of Medical Sciences, University of Kragujevac, 34000 Kragujevac, Serbia; zormil67@gmail.com (Z.M.); nemanjajovicic.kg@gmail.com (N.J.); marina84kv@gmail.com (M.M.K.); 9Department of Radiology, Faculty of Medical Sciences, University of Kragujevac, 34000 Kragujevac, Serbia; 10Department of Dentistry, Faculty of Medical Sciences, University of Kragujevac, 34000 Kragujevac, Serbia; suzanazivanovic91@yahoo.com; 11Laboratory for Bioengineering, Institute of Information Technologies Kragujevac, Department of Natural and Mathematical Sciences, University of Kragujevac, 34000 Kragujevac, Serbia; nevena_milivojevic@live.com; 12Department of Human Pathology, First Moscow State Medical University I.M. Sechenov, 119991 Moscow, Russia; bolevich2011@yandex.ru

**Keywords:** ADMSCs, HBO, osteoarthritis, monoiodoacetate, rats

## Abstract

The beneficial effects of HBO in inflammatory processes make it an attractive type of treatment for chronic arthritis. In addition, the effects of combination therapy based on adipose stem cells and HBO on OA progression have not been fully investigated. The current study explored the efficacy of intra-articular injection of allogeneic adipose-derived mesenchymal stem cells (ADMSCs) combined with hyperbaric oxygenation treatment (HBO) in a rat osteoarthritis (OA) model. The rat OA model was induced by intra-articular injection of monoiodoacetate (MIA) and 7 days after application of MIA rats were divided into five groups: healthy control (CTRL), osteoarthritis (OA), ADMSCs (ADS), the HBO+ADS^21day^ and HBO+ADS^28day^ groups. A single dose of 1 × 10^6^ allogeneic ADMSCs suspended in sterile saline was injected into the knee joint alone or in combination with HBO treatment. Rats were sacrificed at 3 or 4 weeks after MIA injection. Treatment outcomes were evaluated by radiographic, morphological and histological analysis and by specific staining of articular cartilage. We also measured the level of inflammatory and pro/antioxidative markers. We confirmed that combined treatment of ADMSCs and HBO significantly improved the regeneration of cartilage in the knee joint. Rtg score of knee joint damage was significantly decreased in the HBO+ADS^21day^ and HBO+ADS^28day^ groups compared to the OA. However, the positive effect in the HBO+ADS^28day^ group was greater than the HBO+ADS^21day^ group. The articular cartilage was relatively normal in the HBO+ADS^28day^ group, but moderate degeneration was observed in the HBO+ADS^21day^ compared to the OA group. These findings are in line with the histopathological results. A significantly lower level of O_2_^−^. was observed in the HBO+ADS^28day^ group but a higher NO level compared to the HBO+ADS^21day^ group. Moreover, in the HBO+ADS^28day^ group significantly higher concentrations of IL-10 were observed but there was no significant difference in proinflammatory cytokine in serum samples. These results indicate that a single intra-articular injection of allogeneic ADMSCs combined with HBO efficiently attenuated OA progression after 28 days with greater therapeutic effect compared to alone ADMSCs or after 3 weeks of combined treatment. Combined treatment might be an effective treatment for OA in humans.

## 1. Introduction

Osteoarthritis (OA) is a chronic inflammation of the joints characterized by progressive cartilage destruction, subchondral bone sclerosis, formation of marginal osteophytes, and changes in synovial fluid composition [1,2]. The clinical presentation of OA involves chronic pain, morbidity, and disability in elderly people [3]. One of the most commonly affected joints is the knee joint (gonarthritis, GA) which requires long-term and complex treatment with an often partially successful outcome. Currently available treatments for OA include weight control, exercise, and pharmacological approaches, which typically consist of intra-articularly injected viscoelastic supplements and analgesic therapies containing acetaminophen, salicylates, and nonsteroidal anti-inflammatory drugs [1,4]. The standard pharmacological treatment includes agents for control of pain and inflammation (non-steroidal anti-inflammatory drugs, analgesics including opioids, intraarticular corticosteroids] and the group of symptomatic slow acting drugs for OA such as glucosamine sulfate, chondroitin sulfate, diacerein, unsaponifiables extract of soybean and avocado administered orally and intraarticular hyaluronic acid [5]. However, these therapeutic strategies are effective only for a short time and focus only on the temporary relief of symptoms, but not on the pathogenesis of the disease. Therefore, there are medical needs for permanent treatments that can modify the course of the disease.

Recent studies implied that progression of OA is closely related to increased oxidative stress and inflammation. Generally, under conditions of excessive reactive oxygen species (ROS) formation which overcomes the antioxidant defense mechanisms oxidative stress gets started. Under this pathological state, increased ROS production that are usually released from synoviocytes, chondrocytes, and osteoblasts induce proteolytic enzyme release which finally results in degradation of cellular matrix and chondrocyte apoptosis. Due to their well-known chemical properties, ROS has a strong potential to mediate and intensify the characteristic join degradation which makes them a key factor for OA development and inflammatory transformation of joint tissues [6,7]. Allogeneic adipose-derived mesenchymal stem cells (ADMSCs] have the capacity to self-renew and differentiate into various connective tissue cells, including chondrocytes [8]. ADMSCs are found in large numbers in adipose tissue and are easily obtained by liposuction with minimal donor site morbidity [9]. About 1 ± 10% of the nucleated cells in adipose tissue are thought to be ADMSCs, while only 0.0001 ± 0.01% of the nucleated cells in the bone marrow are stem cells [8]. The results of various animal studies and clinical trials have shown promising results in OA therapy following intra-articular administration of ADMSCs [10,11,12]. It has recently been shown that local application of ADMSCs to the knee joint in the early phase of experimental osteoarthritis inhibits chondrocyte autophagy and largely attenuates osteoarthritis [13]. Although there have been promising advances toward the clinical use of ADMSCs, a number of problems have arisen in their application including isolation, method of application, homing, and cell survival, which requires further research in order to have a potential therapeutic benefit of ADMSCs treatment [14].

Cell therapy and tissue engineering have become increasingly common alternative treatments for cartilage defects [15]. Intra-articular injections may include administration of stem cells collected from different sources, platelet-rich plasma (PRP), hyaluronan preparations, and ozone [16]. Mesenchymal stem cells (MSC) are superior to others due to: their self-renewal ability; being essential for normal turnover and maintenance of cartilage; being capable to migrate to the damaged area of cartilage, and having the ability to induce chondrocyte proliferation and extracellular matrix (ECM) synthesis [15].

In addition to soluble mediators such as growth factors, environmental factors may play an important role in regulating stem cell growth and differentiation. The results of some studies indicate that the low concentration of oxygen (5%), in which ADMSCs are grown, can significantly increase the synthesis of collagen and sulfated glycosaminoglycans from ADMSCs, but at the same time inhibit the proliferation of these cells. These findings suggest that oxygen concentration may play an important role in regulating the balance between cell proliferation and their biosynthetic activity. The potential use of oxygen concentration as a “metabolic switch” has significant implications both in vivo and in vitro in the context of tissue engineering [17].

Therefore, in the treatment of OA, in addition to stem cells (or in combination with them), hyperbaric oxygenation (HBO) is mentioned as a potentially effective adjuvant therapeutic procedure. The beneficial effects of HBO on inflammatory processes make it an attractive type of treatment for chronic arthritis [18]. Until now, HBO therapy has been used for the treatment of numerous orthopedic diseases including soft tissue infections, acute traumatic ischemia, crushing, compartment syndrome, problematic wounds, refractory osteomyelitis, osteonecrosis, sports disorders, disease disorders and injuries [19]. However, in the literature, there is a limited number of preclinical studies which show the positive effects of HBO therapy on cartilage tissue [20,21,22]. However, Yılmaz et al. have shown that systemic medical O_3_ application was more effective than HBO therapy and may reduce the development of cartilage damage and prevent OA formation [23]. Recently published studies have shown that exposure of osteogenic-differentiating MSCs to HBO under in vitro simulated inflammatory conditions enhances differentiation towards the osteogenic phenotype, providing evidence of the potential application of HBO in all those processes requiring bone regeneration [24]. A step further in the application of MSC and HBO as a co-treatment is shown by numerous clinical studies. In the therapy of certain diseases such as multiple sclerosis, ALS, sports injuries and stroke, there are no studies on the combined treatment with ADMSCs and HBO in OA knee joint regeneration [25].

In addition, the effects of combination therapy based on adipose stem cells and HBO on OA progression have not been fully investigated. Based on all the above, it is assumed that they could have an even more positive cumulative effect. In this sense, we are of the opinion that the creation of an animal model of OA would be of interest for studying the pathophysiology of OA and would help in the development of therapeutic agents and biological markers for diagnosing and predicting disease.

Our hypothesis was that HBO treatment in conjunction with stem-cell therapy, would improve oxygenation in the rat’s knee joint and leading to joint regeneration.

Using a rat model of osteoarthritis, induced by monoiodoacetate (MIA), followed by HBO therapy and local application of ADMSCs to the knee joint, we have demonstrated significant attenuation of the disease progression.

## 2. Results

### 2.1. Severity of Arthritis–Measured by Diameter of Knee

Our results showed that OA induction in all experimental groups led to slight increase in knee diameter, (respectively, OA 14.91 ± 0.96; ADS 14.68 ± 0.75; HBO+ADS^21day^ 14.43 ± 0.55; HBO+ADS^28day^ 14.30 ± 0.55) compared to CTRL group (12.94 ± 0.82) but without any statistical differences (*p* > 0.05) (Figure 1).

### 2.2. Radiography of OA

The analysis of radiological images showed that the OA group showed progressive changes to the knee joint during the experimental period, whereas the CTRL group maintained a normal appearance. Application of ADMSCs stand-alone showed a mild degree of improvement and healing compared to the group without therapeutic intervention (OA). On the other hand, there was a significant difference in the recovery rate with combined treatment (HBO+ADS) compared to the healthy control group and OA group of animals, especially in the group HBO+ADS^28day^ (Figure 2A,B).

### 2.3. Effect of ADMSCs and HBO on the Repair of Cartilage Defects

In order to define and quantify the degree of degenerative changes in cartilage as well as inflammatory changes in the joint, we used HE and Safranin O staining. The images were captured with a light microscope (Olympus BX51, Tokyo, Japan) equipped with a digital camera. The results obtained from the pathohistological analysis largely coincide with the radiological results. Pathohistological analysis showed a difference between all experimental groups compared to the group without therapeutic intervention (OA), model with combined treatment (HBO+ADS^28day^) is the one that presents more difference.

In the OA group, HE staining clearly showed signs of articular cartilage degeneration. In the surface layer of cartilage, there was mainly one chondrocyte in the lacunae. In the deeper layers of cartilage, a partial lack of chondrocytes in the lacunae were observed. Chondrocytes with signs of degenerative changes and hyperchromatic nuclei were observed (Figure 3A). Additionally, Safranin O-positive hyaline-like cartilage was almost completely absent in the group with OA compared to the CTRL group, which clearly indicates the progression of degenerative changes (Figure 3B).

At 21 days, histologic assessment demonstrated that application of ADMSCs stand-alone leads to partial recovery of knee joint with preservation of cartilage smooth surface with normal thickness and matrix intensity. A higher number of chondrocytes is observed in the cartilage, with partially preserved morphology (Figure 3A). Additionally, Safranin O–positive hyaline-like cartilage was less prominent in the ADS group, compared to the OA group (Figure 3B).

In the group with the combined treatment (HBO+ADS^21day^), a greater thickness of cartilage was clearly observed in relation to the other examined groups. The cartilage surface was smooth. In all three zones of cartilage, mostly single chondrocytes were observed in lacunaes with preserved cellular morphology (Figure 3A). Additionally, the cartilage showed more intense Safranin O staining, which spreads on the surface, middle and deep zone of the cartilage, indicating a higher content of glycosaminoglycans (GAG). Based on the presented results, it can be concluded that the combined treatment accelerates the process of cartilage regeneration, in particular 28 days after treatment (HBO+ADS^28day^) (Figure 3B).

### 2.4. Parameters of Oxidative Stress

#### 2.4.1. Prooxidative Markers

The concentration of O_2_^−^ was significantly increased in both treated groups compared to OA and CTRL groups. However, there were no statistical differences in the level of this prooxidative marker between ADS and HBO+ADS groups as well as between OA and CTRL. Additionally, combined HBO and ADS treatment caused significant reduction in H_2_O_2_ level in comparison to the OA and ADS groups but already stays significantly increased compared to CTRL group. Additionally, the levels of this prooxidant were significantly increased in OA and ADS groups compared to healthy control rats Nitrites levels were not significantly changed in all experimental groups but were significantly increased compared to CTRL group. Additionally, the most prominent change was observed in TBARS value in OA group compared to other groups. While significantly increased level of TBARS was seen in OA group, treatment protocol restored the value of this marker to those observed in CTRL group. Finally, different treatment protocol did not lead to significant changes in the level of this prooxidative marker (Figure 4A).

Following concentrations of pro-oxidative molecules in plasma samples in two different points of interest, a significantly lower level of O_2_^−^ was observed after 28 days of OA induction compared to its value after 21 days of disease induction. On the other hand, there were no statistical differences in the H_2_O_2_ and TBARS levels between the HBO+ADS^21day^ and HBO+ADS^28day^ groups. However, nitrites were significantly increased after 4 weeks of OA induction compared to their value in the HBO+ADS^21day^ group (Figure 4B).

#### 2.4.2. Antioxidative Enzymes

Applied treatment protocol caused significantly increased activity of antioxidative enzymes compared to OA group. Additionally, single ADS treatment caused remarkably higher GSH and CAT activity compared to OA group but oppositely led to the lower SOD activity. Moreover, it was observed that OA induction led to significant reduction in the activity of antioxidant enzymes compared to healthy control rats. (Figure 5A).

On the other hand, there were no statistical differences in the antioxidative enzyme activity between HBO+ADS^21day^ and HBO+ADS^28day^ groups (Figure 5B).

### 2.5. ELISA of Cytokines

We next examined the levels of TNF-α, IL-6, IL-10 and vascular endothelial growth factor (VEGF) in the serum samples of experimental animals after induction of osteoarthritis. Serum concentrations of TNF-α were significantly increased in the sera of ADS and HBO+ADS^21day^ group compared to the CTRL and OA groups of experimental animals. Analysis of serum TNF-α value at two time points showed a higher level of HBO+ADS^21day^ group compared to HBO+ADS^28day^ group, but without a statistically significant difference. Analysis of the values of IL-6 and IL-10, which are a cytokine related to the profile of TH2 and anti-inflammatory response, showed significantly higher serum levels in the ADS and HBO+ADS^21day^ groups compared to the CTRL and OA groups. Additionally, higher serum concentrations of IL-6 were observed in the HBO+ADS^21day^ group compared to the HBO+ADS^28day^ group (without a statistically significant difference), while significantly higher levels of IL-10 were observed in the HBO+ADS^28day^ group. However, analysis of absolute ratios showed that TNF-α/IL-10 and IL-6/IL-10 were higher in the ADS group compared to the HBO+ADS^21day^ group, apparently due to increased serum TNF-α and IL-6 in serum of each individual rat. Statistically significantly higher levels of VEGF were observed in sera ADS and HBO+ADS^21day^ group compared to the CTRL and OA groups. Examination of serum VEGF values at two different points of interest showed a higher level of VEGF after 28 days of OA induction in the HBO+ADS^28day^ group compared to its value in the HBO+ADS^21day^ group (Figure 6 and Figure 7).

## 3. Discussion

Osteoarthritis is a chronic joint disease characterized by degeneration of the articular cartilage, surrounding bone and chronic pain. The MIA-induced OA animal model used in the present study is the most similar model which simulates pain and biochemical/structural changes associated with human OA. In this study we first used, different therapeutic models of the acute phase of monoiodoacetate-induced animal model of osteoarthritis which were examined and compared. We demonstrated that the application of the combined treatment (HBO+ADS) has the greatest potential for cartilage repair in the chemically induced OA model. The therapy effect was manifested by reduced cartilage degeneration, and induction of anti-inflammatory response. The results of our study demonstrated a significantly increased level of pro-oxidative molecules as well as reduced antioxidant enzyme activity in OA group compared to healthy control rats. Moreover, it was observed that the combined treatment (HBO+ADS) group had the strongest potential to improve antioxidant defense thus preventing oxidative stress occurring in the experimental model of OA. In addition, examining the time-course effects of HBO treatment we did not mention any differences in the duration of applied treatment on the redox status of animals with OA. Main recorded characteristics of ADMSCs treatment on OA in animal studies and clinical trials were their big differentiation capacity, as well as anti-inflammatory effect and regenerative functions [26]. There are also reports about the safety and efficacy of culture expended allogenic ADMSC treatment for OA [27]. The main drawback is in vivo survival rate of ADMSCs alone which is low and limits the therapeutic effect of ADMSCs. In the current study, we first used ADMSCs in combination with HBO to treat knee joint osteoarthritis. Results from our study are in line with in vitro study conducted by D. Wang et al. Their findings suggest that oxygen tension has an important role in regulating the proliferation and metabolism of human adipose-derived adult stem (*h*ADAS) cells as they undergo chondrogenesis, and the exogenous control of oxygen tension may provide a means of increasing the overall accumulation of matrix macromolecules in tissue-engineered cartilage [17]. A previous study found that the positive effect of allogeneic ADSCs was observed as early as 4 weeks after transplantation. Intra-articular injection of culture expanded allogenic ADSCs attenuated cartilage degeneration in an experimental rat OA model without observed adverse effects [27]. In the current study, we sacrificed the animals at 3 and 4 weeks after treatment. We found that ADMSCs injection and HBO treatment after 28 days reversed cartilage degeneration based on both macroscopic and histological evaluations. Combined HBO and ADMSCs treatment decreased the serum levels of TNF-α and IL-6, while serum levels of IL-10 and VEGF were increased. The treatments with ADMSCs alone also exerted a protective effect on OA, but neither effect was better than the combined HBO and ADMSCs treatment. Research by Harnanik et al. shown that HBO treatment has effects on the polarization of Th17 to Treg through a decrease in expression of HIF-1α in mice with antigen and collagen-induced arthritis (ACIA). The conclusion of the aforementioned authors is that HBO treatment can be good supporting therapy for RA in combination with drug therapy [28]. Data from the literature indicate the therapeutic pathway of ADMSCs in OA by paracrine action with the secretion of anti-inflammatory factors [29]. Zhou et al. have shown that ADMSCs alleviated OA and inhibited cartilage degeneration through reduced secretion of proinflammatory cytokines and protected against apoptosis through autophagy inducing. Proinflammatory cytokines are suppressed when ADMSCs are treated in vitro and in vivo, indicating that ADMSCs have a paracrine effect and may secrete multiple anti-inflammatory and growth factors. It has also been shown that ADMSCs treatment attenuated FGF-1 expression, which may also be responsible for the regenerative function of ADMSCs. Additionally, ADMSCs play an important role in modulating FGFR-1, DDR-2, Wnt, *p*-Smad1/Smad, *p*-CAMK II/CAMKII, and *p*-AKT/AKT signaling. ADMSCs’ function could be related to multiple signaling pathways [13,29].

Another aspect of our study involved examining if alone ADMSCs or combined with HBO treatment can change systemic concentration of pro-oxidative molecules. The results showed significantly lower levels of all pro-oxidative markers in these two groups compared to OA, except of elevated level of superoxide anion radical in treated groups. This finding could be explained through accumulating evidence that demonstrated a modest increase in intracellular ROS during routine HBO treatment can modulate signaling pathways, induce the expression of cytoprotective proteins and enhance cellular tolerance of harmful stimuli [30]. Although increased ROS production was noticed after HBO treatment, the high oxygen concentration paradoxically led to an antioxidant environment in plasma by increasing the activity of antioxidant enzymes [31]. Additionally, different studies showed an increase in total antioxidant capacity after HBO exposure [31,32,33]. This is in line with the results of our study demonstrating significantly higher activity of each antioxidant enzyme in treated groups compared to OA. Moreover, the special fact of this study is to emphasize that there were no changes in antioxidant defense during two different time intervals. The therapeutic use of HBO may be reflected in increased ROS production which vice versa improves the regulation of antioxidant enzyme activity of tissues.

In summary, a single intraarticular injection of allogeneic ADMSCs and HBO treatment attenuated OA progression in a rat OA model. The therapeutic effect of ADMSCs-HBO was significantly higher than ADMSCs alone. The potential biological effects of HBO and ADMSCs, as well as the actual action mechanism of ADMSCs treatment on OA, remain to be elucidated.

## 4. Materials and Methods

### 4.1. Induction of the Osteoarthritis Model

Intra-articular injection of MIA (cat. #I2512; Sigma, St. Louis, MO, USA) was used for OA induction in this study, as previously described [34]. Among three tested doses of MIA (1 mg/kg, 1.5 mg/kg and 2 mg/kg), the dose of 2 mg/kg was selected as the most efficient for OA induction and was dissolved in 50 μL of the sterile saline. After a short-term ketamine/xylazine narcosis (ketamine 100 mg/kg; xylazine 10 mg/kg), right knee of the hind leg was shaved and disinfected and 50 μL of MIA was injected into the medial side of the patellar ligament using 30-gauge, 0.5-inch needle. 

### 4.2. Cell Culture

For cell culture we used Wistar rat adipose-derived mesenchymal stem cells (ADMSCs) purchased from Cyagen Biosciences Inc., Santa Clara, California, USA (Lot Number: 120824L01). The ADMSCs cells were maintained in MesenPRORS^TM^ medium (Gibco^TM^, ThermoFisher Scientific, Waltham, MA, USA) supplemented with 10% fetal bovine serum (Gibco, Grand Island, NY, USA), 100 U/mL penicillin (Sigma-Aldrich, Taufkirchen, Germany), 100 μg/mL streptomycin (Sigma-Aldrich, Taufkirchen, Germany), 2 mM L-glutamine (Sigma-Aldrich, Taufkirchen, Germany) and 1 mmol/l non-essential amino acids (Capricorn Scientific GmbH, Ebsdorfergrund, Germany). Cells were cultivated at 37 °C in an atmosphere of 5% CO_2_ and absolute humidity. The culture medium was completely replaced every 3 days until cells reached 80% confluence, after which they were passaged and replated. After the fourth passage (P4), cells were separated from the bottom of the flask by short-acting 0.25% trypsin–EDTA (Gibco, Grand Island, NY, USA). The cells were then resuspended in 10 mL of MesenPRO RS^TM^ medium to neutralize trypsin and prevent cell damage and then centrifuged at 1500 RPMI for 10 min. Cells were resuspended in 1 mL of MesenPRO RS^TM^ medium and cell viability was determined using trypan blue staining and only cell suspensions with viability greater than 95% were used.

### 4.3. Experimental Animals and Animal Care

A total of 40 male *Wistar albino* rats (8 per group) were used for the present study (8 weeks old; 250 ± 50 g body mass). The animals were housed in an air-conditioned room (22 ± 2 °C) with an established photo period of 12 h light/day and had free access to food and tap water ad libitum. For research purposes, all experimental animals were obtained from the Military Medical Academy, Belgrade, Serbia. Initially, rats were randomly divided into 4 groups (8 animals per group) according to the applied protocol as follows:Healthy control rats (CTRL)Rats with induced OA without therapeutic intervention, untreated (OA)Rats with induced OA subjected to ADMSCs treatment (ADS)Rats with induced OA subjected to single ADMSCs injection and HBO treatment for 14 days (HBO+ADS^21day^)

The second part of this study included examination of the time-course effects of HBO treatment during 14 consecutive days on OA progression after single ADS injection. In that sense, 8 more animals were included in this study and were subjected to HBO treatment during 14 days after OA induction. Animals from this group were sacrificed after 28 days of OA induction (HBO+ADS^28day^ group) in order to compare cartilage damage to rats from HBO+ADS^21day^ group [35,36].

The experimental protocol was approved by the Ethics Committee for Experimental Animal Well Being of the Faculty of Medical Sciences of the University of Kragujevac, Serbia (No: 01–71921), and followed the Guidelines or the Care and Use of Laboratory Animals.

### 4.4. ADMSCs Treatment Protocol

At 7th day of OA induction, each rat from ADS group received IA injection of 1 × 10^6^ ADMSCs suspended in 60 μL of sterile saline according to previously described protocol [37].

### 4.5. HBO Treatment

A specially constructed hyperbaric pressure chamber (HYB-C 300, Maribor, Slovenia) was used for the experiment. Rats were exposed to 100% oxygen once daily for 14 days for 60 min at a pressure of 2.5 ATM and a flow rate of 7 L/min ± 10%. To avoid the effects of diurnal rhythm variations, hyperbaric oxygenation always began at the same time. 

### 4.6. Radiographic Analysis of OA

The right hind leg knee of all animals was radiographed using an X-ray unit (Sirona Dental Systems Bensheim, Germany) on 21st and 28th days after OA induction. Exposure time was set to 0.12 s (70 kVp, 7 mA). Custom equipment was used to place the leg at the same angle and distance from the sensor to achieve uniformity during radiography. The observed changes were evaluated semi-quantitatively. The tissue samples were further used for pathohistological analysis. X-ray images were classified based on the Kellgren and Lawrence system [38] as grade 0 (none: no radiographic features of OA), grade 1 (doubtful: doubtful joint space narrowing (JSN) and possible osteophyte), grade 2 (minimal: the presence of definite osteophytes and possible JSN), grade 3 (moderate: multiple osteophytes, definite JSN, sclerosis, possible bony deformity), and grade 4 (severe: large osteophytes, marked JSN, severe sclerosis and definite bony deformity).

### 4.7. Knee Diameter

Knee diameter was measured using a calibrated digital calliper (World Precision Instruments, Stevenage, UK) in millimeters (mm) to evaluate the developmental stages of OA on days 0 (pre-MIA injection), 21, and 28 (post-injection).

### 4.8. Histological Preparation

Immediately after the animals were sacrificed, the right hindlimb was disarticulated at the hip joint, fixed in 10% neutral-buffered formalin for 72 h and dehydrated in alcohol series. The fixed samples were embedded in paraffin blocks, cut into 8 μm thick sections and stained separately with hematoxylin and eosin (HE) and 0.1% Safranin O fast green for 5 min [39]. Afterwards, the slides were sequentially dehydrated in 70%, 80%, 90%, and 100% ethanol and finally cleared in xylene. A light microscope with an associated digital camera (BX51, Olympus, Japan) was used to assess the histopathological characteristics of articular cartilage. Microscopy was performed in blinded fashion by two investigators (Z.M. and M.M.K.).

### 4.9. Markers of Oxidative Stress and Inflammation

After short ketamine/xylazine narcosis, animals were sacrificed, and blood samples were collected. In serum samples, we determined the levels of cytokines (TNF-α, IL-6, IL-10, and VEGF) using enzyme-linked immunosorbent assays (ELISA) according to a predetermined protocol (R&D Systems, Minneapolis, MN, USA).

Plasma samples were used for measurement of prooxidative markers including superoxide anion radical (O_2_^−^), hydrogen peroxide (H_2_O_2_), nitric oxide (NO^−^) and the index of lipid peroxidation measured as TBARS (TBARS). In the lysate, the activity of reduced glutathione (GSH) as a non-enzymatic antioxidant was measured as well as the activities of enzymatic defense system including catalase (CAT) and superoxide dismutase (SOD).

The level of the O_2_^−^ was measured via reaction of nitro blue tetrazolium in TRIS buffer with the plasma samples at 530 nm. Distilled water solution was served as a blank probe [40]. Determination of H_2_O_2_ levels in plasma samples was based on the oxidation of phenol red by hydrogen peroxide in a reaction catalyzed by horseradish peroxidase (HRPO) [41]. Using the distilled water as a blank probe, measurement was performed at 610 nm. Using the previously described protocol, an indirect method for monitoring NO by determining nitrate (NO_3_^−^) and nitrite (NO_2_^−^) was performed in this study [42]. A total of 0.5 mL of plasma was precipitated with 200 μL of 30% sulfosalicylic acid, vortexed for 30 min, and centrifuged at 3000× *g*. Equal volumes of the supernatant and Griess reagent, containing 1% sulphanilamide in 5% phosphoric acid/0.1% naphthalene ethylenediamine dihydrochloride, were added and incubated for 10 min in the dark and measured at 543 nm. In order to determine the degree of lipid peroxidation in the plasma samples, we measured TBARS using 1% thiobarbituric acid in 0.05 NaOH which was incubated with plasma at 100 °C for 15 min and measured at 530 nm. Distilled water was used as a blank probe [43]. The level of GSH was based on the determination of GSH oxidation with 5.5-dithio-bis-6.2-nitrobenzoic acid using the method of Beutler [44]. Activity of the CAT was monitored according to Aebi. Lysates were diluted with distilled water (1:7 *v*/*v*) and treated with chloroform-ethanol (0.6:1 *v*/*v*) to remove hemoglobin. Then, 50 μL of CAT buffer, 100 μL of sample and 1 mL of 10 mM H_2_O_2_ were added to the samples and measured at 360 nm [45]. Using the epinephrine method of Beutler, we detected the activity of SOD. A total of 100 μL of lysate and 1 mL of carbonate buffer were mixed, 100 μL of epinephrine was added and measured at 470 nm [46].

### 4.10. Statistical Analysis

Statistical analysis was conducted using SPSS 22.0 statistical package. Data are presented as mean ± standard error (SE) or as medians (5th, 25th, 75th and 95th percentiles). The normality of the distribution of the data was determined using the Shapiro–Wilk test. If the values had the correct distribution, the parametric Student’s *t* test was used, while the incorrect distribution was compared using the nonparametric Mann–Whitney test. In fact, we performed the one-way ANOVA and Bonferroni test for multiple comparisons of parameters of systemic redox status which were determined in blood samples in one time point between four groups. Additionally, in our study data generated from time-course measurements such as changes in parameters of oxidative stress over the time (after 21 or 28 days) were analyzed using two-way ANOVA followed by Bonferroni posttest to account for the two variables of time and treatment. The accepted statistical significance values were *p* < 0.05.

## 5. Conclusions

In conclusion, we demonstrated that combined use of intraarticularly application adipose-derived mesenchymal stem cells (ADMSCs) and HBO treatment, is capable of downregulating inflammatory factors and prooxidative factors, alleviating knee osteoarthritis and finally significantly attenuating disease progression. Considering that reconstructive cell therapy and HBO treatment are becoming recognized in clinics, particularly in orthopedics, our study justified the potential use of combined ADSMCs and HBO treatment as a novel therapeutic modality to impede the pathologic course of knee OA.

## Figures and Tables

**Figure 1 ijms-23-07695-f001:**
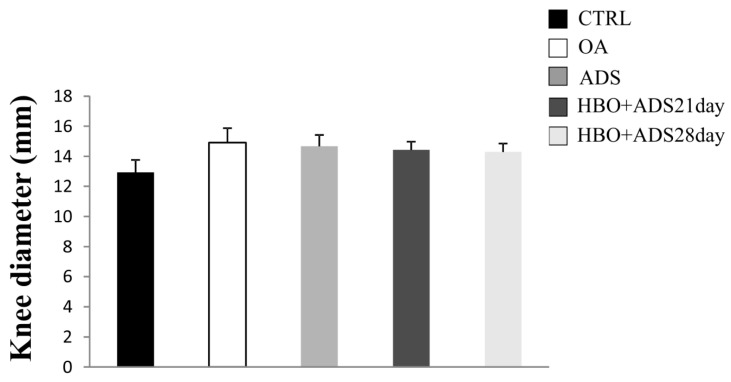
Changes in knee diameter in CTRL, OA, ADS, HBO+ADS^21day^ and HBO+ADS^28day^ groups. Values are presented as means ± mean standard error (SE).

**Figure 2 ijms-23-07695-f002:**
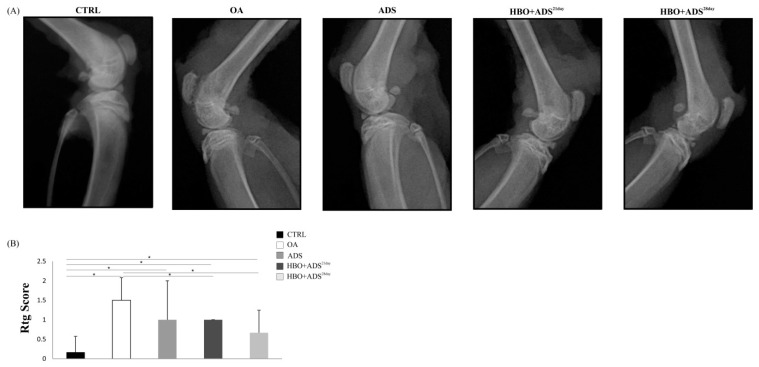
(**A**) Representative radiography images and (**B**) graphic analysis of knee joints in CTRL, OA, ADS, HBO+ADS^21day^ and HBO+ADS^28day^ groups. Values are presented as means ± mean standard error (SE). * *p* < 0.05 indicates statistically significant differences between groups.

**Figure 3 ijms-23-07695-f003:**
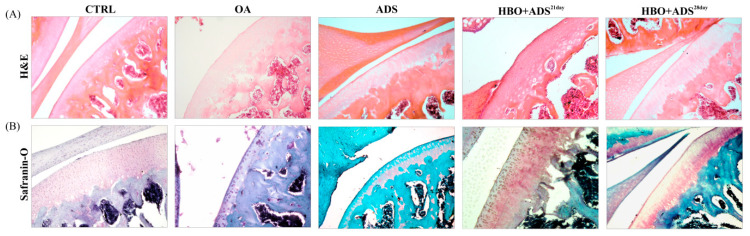
Representative images of knee joint sections stained with (**A**) hematoxylin/eosin (H&E) and (**B**) Safranin O-fast green (Safranin O) in CTRL, OA, ADS, HBO+ADS^21day^ and HBO+ADS^28day^ groups. Original magnification 100×.

**Figure 4 ijms-23-07695-f004:**
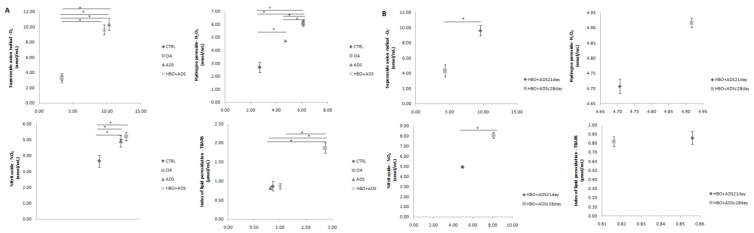
(**A**) Changes in the value of prooxidative markers in plasma samples between CTRL, OA, ADS and HBO+ADS groups measured as superoxide anion radical (O_2_^−^), hydrogen peroxide (H_2_O_2_), nitrites (NO_2_^−^) and index of lipid peroxidation (TBARS). (**B**) Changes in the value of prooxidative markers in plasma samples between HBO+ADS^21days^ and HBO+ADS^28days^ groups measured as superoxide anion radical (O_2_^−^), hydrogen peroxide (H_2_O_2_), nitrites (NO_2_^−^) and index of lipid peroxidation (TBARS). Values are presented as means ± mean standard error (SE). * *p* < 0.05 indicates statistically significant differences between groups.

**Figure 5 ijms-23-07695-f005:**
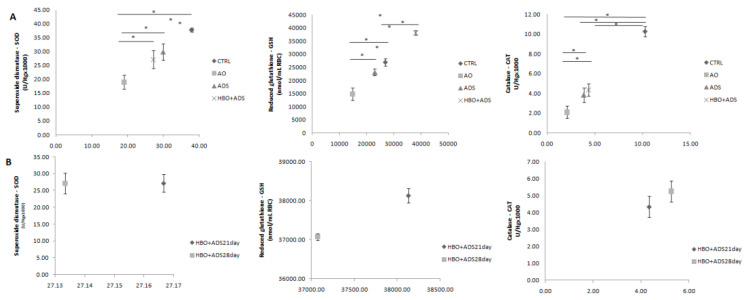
(**A**) Changes between CTRL, OA, ADS and HBO+ADS groups in the antioxidant enzyme activity in samples of erythrocyte lysate measured as superoxide dismutase (SOD), reduced glutathione (GSH) and catalase (CAT). (**B**) Changes between HBO+ADS^21days^ and HBO+ADS^28days^ groups in the antioxidant enzyme activity in samples of erythrocyte lysate measured as superoxide dismutase (SOD), reduced glutathione (GSH) and catalase (CAT). Values are presented as means ± mean standard error (SE). * *p* < 0.05 indicates statistically significant differences between groups.

**Figure 6 ijms-23-07695-f006:**
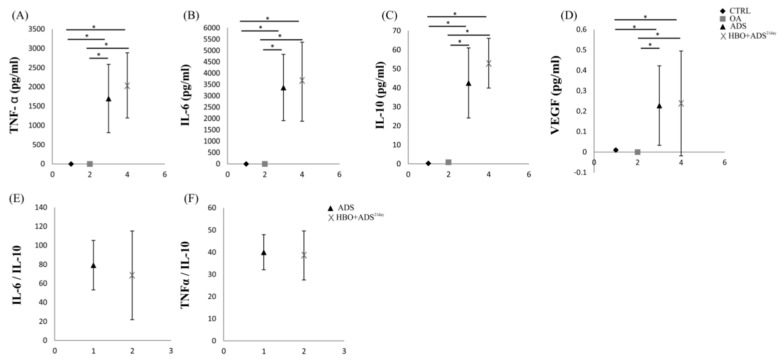
Changes in cytokine serum levels measured as (**A**) TNF-α, (**B**) IL-6, (**C**) IL-10 and (**D**) VEGF between CTRL, OA, ADS and HBO+ADS^21days^ groups. Changes in (**E**) IL-6/IL-10 ratio and (**F**) TNF-α/IL-10 ratio between ADS and HBO+ADS^21day^ groups. * *p* < 0.05 indicates statistical significant differences between groups.

**Figure 7 ijms-23-07695-f007:**
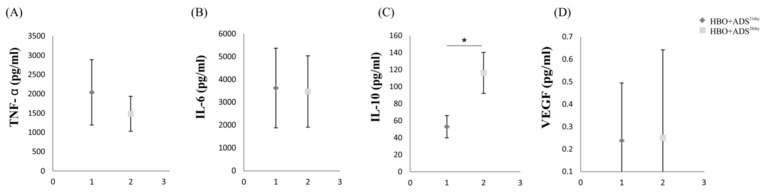
Changes in cytokine serum levels measured as (**A**) TNF-α, (**B**) IL-6, (**C**) IL-10 and (**D**) VEGF between HBO+ADS^21day^ and HBO+ADS^28day^ groups. * *p* < 0.05 indicates statistical significant differences between groups.

## Data Availability

All data are available from the author’s upon request.

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
