# Peer review of "Effects of Combined Allogenic Adipose Stem Cells and Hyperbaric Oxygenation Treatment on Pathogenesis of Osteoarthritis in Knee Joint Induced by Monoiodoacetate"

_ijms, 2022, doi:10.3390/ijms23147695_

Round 1
Author Response
Kragujevac, 07/04/2022
Dear Ms. Adela Liu,
Many thanks for your patience and the evaluation of our manuscript entitled “Effects of combined allogenic adipose stem cells and hyperbaric oxygenation treatment on pathogenesis of osteoarthritis in knee joint induced by monoiodoacetate” (jcm- 1779090).
We are thankful for all critical suggestions and comments raised by reviewers and hope that we were able to improve the quality of our manuscript. Please find below our detailed responses and explanations. Please note that all changes are highlighted in red and marked up using the “Track Changes” function.
To Review 1.
We are very thankful to Review 1 since the reviewer invested a plethora of time and efforts to improve the quality of our manuscript raising more than constructive questions and suggestions. She/he stated that:
„The manuscript fits the International journal of molecular sciences and will certainly attract attention of even broader audience, especially because it deals with very important topics in a very modern way. Therefore, it deserves considering for publication, but after some major and minor revision, which should include.“
MAJOR ISSUES
- „The discussion section is a description of the results rather than an explanation of the changes. It should definitely be expanded (mechanisms of action).“
Response: Thank You for your suggestion. This is a very helpful advice. Based on Your comments, we have enriched the discussion with the possible mechanisms of adipose-derived mesenchymal stem cells (ADMSCs) action. Also, according to the Reviewer's 2 suggestion mechanisms of actions hyperbaric oxygenation, alone and in combination with ADMSCs are added in Introduction.
Discussion: Data from the literature indicate the therapeutic pathway of ADMSCs in OA by paracrine action with secretion of anti-inflammatory factors (Wu, L.; Cai, X.; Zhang, S.; Karperien, M.; Lin, Y. Regeneration of articular cartilage by adipose tissue derived mesenchymal stem cells: perspectives from stem cell biology and molecular medicine. J Cellular Physiol. 2013, 228, 938‐944.).
Zhou et al. have shown that ADMSCs alleviated OA and inhibited cartilage degeneration through reduced secretion of proinflammatory cytokines and protected against apoptosis through autophagy inducing. Proinflammatory cytokines are suppressed when ADMSCs are treated in vitro and in vivo, indicating that ADMSCs have a paracrine effect and may secrete multiple anti-inflammatory and growth factors. It has also been shown that ADMSCs treatment attenuated FGF‐1 expression, which may also be responsible for the regenerative function of ADMSCs. Additionally ADMSCs play an important role in modulating FGFR‐1, DDR‐2, Wnt, p‐Smad1/Smad, p‐ CAMK II /CAMKII, and p‐AKT/AKT signaling. ADMSCs’ function could be related to multiple signaling pathway (Zhou, J.; Wang, Y.; Liu, Y.; Zeng, H.; Xu, H.; Lian, F. Adipose derived mesenchymal stem cells alleviated osteoarthritis and chondrocyte apoptosis through autophagy inducing. J Cell Biochem. 2019, 120, 2198-2212.).
- „Some of the results are not described in the discussion.“
Response: Thank You for this kind comment. We reformulated discussion section. Please see page 10.
- „The conclusion is definitely too short and does not bring anything new, there are no details and explanations.“
Response: Thank You for the sharp observation. In the new version of the manuscript, we modified conclusion section according your suggestions.
New conclusion: In conclusion, we demonstrated that combined use of intraarticularly application adipose-derived mesenchymal stem cells (ADMSCs) and HBO treatment, is capable of downregulating inflammatory factors and prooksidativ factors, alleviating knee osteoarthritis and finally significantly attenuates disease progression. Considering that reconstructive cell therapy and HBO tretmant are becoming recognized in clinics, particularly in orthopedics, our study justified the potential use of combined ADSMCs and HBO treatment as novel therapeutic modality to impede the pathologic course of knee OA.
- „There is nothing about oxidative stress and inflammation in introduction part.“
Response: We absolutely agree with Your comment. Please see the new paragraph in the introduction referring the role of oxidative stress and inflammation in the OA conditions.
- „Explanation about the planned experiment why these specific days were chosen, perhaps by reference to the literature.“
Response: Thank You for your kind suggestion. The experimental protocol was performed according to the literature data and references were included in the section Materials and methods (Sampath, S.J.P.; Kotikalapudi, N.; Venkatesan, V. A novel therapeutic combination of mesenchymal stem cells and stigmasterol to attenuate osteoarthritis in rodent model system-a proof of concept study. Stem Cell Investig. 2021, 8, 5.; Bar-Yehuda, S.; Rath-Wolfson, L.; Del Valle, L.; Ochaion, A.; Cohen, S.; Patoka, R.; Zozulya, G.; Barer, F.; Atar, E.; Piña-Oviedo, S.; Perez-Liz, G.; Castel, D.; Fishman, P. Induction of an antiinflammatory effect and prevention of cartilage damage in rat knee osteoarthritis by CF101 treatment. Arthritis Rheum. 2009, 60, 3061-3071.)
MINOR CORRECTIONS
- „This sentence in the abstract is out of place ,,We confirmed that combined treatment of ADMSCs and HBO significantly.“
Response: We reformulated the abstract according to Your comment. Please see page 1.
- „Standardize the font.“
Response: Thank You. We standardized the font size throughout the manuscript.
- „Materials and methods section of ,,Markers of oxidative stress’’ maybe change markers of oxidative stress and inflammation. More detail about methods and separate them.“
Response: Thank You for this suggestion. Please see corrected paragraph in Material and methods section.
- „Check statistical significance for example Figure 4. nitrites (NO2-) and hydrogen peroxide H2O2;Figure 5. superoxide dismutase (SOD), GSH.“
Response: Thank You for this kind comment. According to Your proposal, we again preformed statistical analyzes but there were no statistically significant differences in the levels of NO2-, H2O2 and GSH, while statistical differences in the SOD levels was observed between examined groups. We are apologizing for this mistake. Please see this correction in Figure 5. Moreover, we included new group of animals and involved that values in statistical analyzes.

Reviewer 2 Report
In this study, the authors have investigated/reported the effects of allogenic adipose stem cell treatment alone or combined with hyperbaric oxygenation treatment on pathogenesis of osteoarthritis (OA) in in vivo models. The treatment responses were recorded after 21 days or 28 days in separate groups. The experimental data suggests that single intra-articular injection of allogeneic adipose-derived mesenchymal stem cells combined with hyperbaric oxygenation efficiently attenuated the OA progression after 28 days, compared to allogenic adipose-derived mesenchymal stem cells alone. Also, the therapeutic effects of 4 weeks of combined treatment were higher than 3 weeks of combined treatment.
The current study is interesting, and suggests time course of treatment effects. However, the presentation of results needs to be improved, and some issues should be addressed.
My comments are appended below.
1. In the first line of the abstract, authors mentioned allogeneic adipose tissue derived stem cells (ADMSCs). Please include mesenchymal stem cells for the used abbreviation MSCs. Same should be updated in the methods if stem cells were adipose derived mesenchymal stem cells.
2. In the abstract, before writing what was done in the current study e.g., ‘’The current study explored the efficacy of…..’’, please provide a background, and biological problem or knowledge gap, then connect the aim of the study with existing knowledge gap/problem.
3. The following statement needs to be corrected. Knee joint damage was significantly decreased in the ADS+HBO21day and ADS+HBO28day groups compared to the OA. In fac, ADS, ADS+HBO21day and ADS+HBO28day treatments were administered to AO model, thus saying it that treatments were compared to the OA is technically unsound. Authors need to clearly mention something like ADS treated OA, ADS+HBO21day OA, and ADS+HBO28day OA, compared to untreated AO. i.e., all 4 groups all AO, three are treated OA and one is untreated OA control. Then there is a healthy control. Please also mention how was the data compared to healthy controls.
4. In abstract, while authors mention the positive effect: please specify the positive effect.
5. In the introduction, include more literature about existing treatments options for OA, then allogenic stem cell treatments, and hyperbaric oxygenation treatments, then challenges and advantages of allogenic stem cell treatments, and hyperbaric oxygenation treatments and line this up with the aim of current study.
6. In the methods: Please include ethical approval statement, with ethical approval number, and the year approved.
Results:
7. Figure 2 Legends: mention the panel A, and B. Same for Figure 3 and 4, i.e, the panel A, and B should be mentioned in the legends.
8. Figure 7D: How was the statistical difference between TN-alpha of day 21 and day 28. Similarly, how was the statistical difference between VEGFA quantified from day21, and day 28. If it was non-significant mention the legends, or letter ns on the figures.
9. Figure 4 to figure 7: Why authors have not included the data from ADS alone, and untreated OA, an healthy controls. It is mandatory to compare the responses of treated groups with untreated OA, and healthy controls. Thus, I suggest authors to include the controls to draw the proper conclusions of the data.
10. In the legends of all figures, with each panel, please indicate the number of replicates (n =?). and where statistical analysis is applied mention the name of tool or parameter applied.
11. In figure panels showing bar graphs, especially Figure 2B, Figure 4 to 7. please replace the bar charts by dot plots to show the position/distribution of individual biological replicates.
Others:
(i). Please avoid control (CTRL) abbreviation in the abstract, if it was not used afterwards.
(ii). Similarly, in the last part of the abstract: either define the abbreviation IA, or better to write it intra-articular injection instead.
Finally, the text style, founts, and grammar need to be corrected throughout the text.
Author Response
Kragujevac, 07/04/2022
Dear Ms. Adela Liu,
Many thanks for your patience and the evaluation of our manuscript entitled “Effects of combined allogenic adipose stem cells and hyperbaric oxygenation treatment on pathogenesis of osteoarthritis in knee joint induced by monoiodoacetate” (jcm- 1779090).
We are thankful for all critical suggestions and comments raised by reviewers and hope that we were able to improve the quality of our manuscript. Please find below our detailed responses and explanations. Please note that all changes are highlighted in red and marked up using the “Track Changes” function.
To Review 2.
We are very thankful to Review 2 since the reviewer invested a plethora of time and efforts to improve the quality of our manuscript raising more than constructive questions and suggestions. She/he stated that:
„In this study, the authors have investigated/reported the effects of allogenic adipose stem cell treatment alone or combined with hyperbaric oxygenation treatment on pathogenesis of osteoarthritis (OA) in in vivo models. The treatment responses were recorded after 21 days or 28 days in separate groups. The experimental data suggests that single intra-articular injection of allogeneic adipose-derived mesenchymal stem cells combined with hyperbaric oxygenation efficiently attenuated the OA progression after 28 days, compared to allogenic adipose-derived mesenchymal stem cells alone. Also, the therapeutic effects of 4 weeks of combined treatment were higher than 3 weeks of combined treatment. The current study is interesting, and suggests time course of treatment effects. However, the presentation of results needs to be improved, and some issues should be addressed.
My comments are appended below.“
- „In the first line of the abstract, authors mentioned allogeneic adipose tissue derived stem cells (ADMSCs). Please include mesenchymal stem cells for the used abbreviation MSCs. Same should be updated in the methods if stem cells were adipose derived mesenchymal stem cells.“
Response: Thank you for the sharp observation. The labeling of the cell cultures, which were used in the study, was corrected according to the manufacturer (https://www.cyagen.com/us/en/product/wistar-adipose-derived-mesenchymal-stem cells.html), instead allogeneic adipose tissue derived stem cells, we replaced term with allogeneic adipose-derived mesenchymal stem cells (ADMSCs). Please see Abstract, Page 1.
To denote groups and display them in figures, we used the abbreviation ADS for marking adipose-derived mesenchymal stem cells, introduced in the abstract section.
- „In the abstract, before writing what was done in the current study e.g., ‘’The current study explored the efficacy of…..’’, please provide a background, and biological problem or knowledge gap, then connect the aim of the study with existing knowledge gap/problem.“
Response: We are thankful for this useful comment. According to Your suggestion, we have already provided existing knowledge on this topic and connected them with the goal of this study. Please see Abstract, Page 1.
- „The following statement needs to be corrected. Knee joint damage was significantly decreased in the ADS+HBO21dayand ADS+HBO28day groups compared to the OA. In fac, ADS, ADS+HBO21day and ADS+HBO28day treatments were administered to AO model, thus saying it that treatments were compared to the OA is technically unsound. Authors need to clearly mention something like ADS treated OA, ADS+HBO21day OA, and ADS+HBO28dayOA, compared to untreated AO. i.e., all 4 groups all AO, three are treated OA and one is untreated OA control. Then there is a healthy control. Please also mention how was the data compared to healthy controls.“
Response: Thank You for this suggestion. In the Material and methods section healthy control rats was marked as CTRL, while the untreated group of experimental animals was marked as untreated, OA.
- „In abstract, while authors mention the positive effect: please specify the positive effect.“
Response: Thank You for this suggestion. Please see Abstract, Page 1. We confirmed that combined treatment of ADMSCs and HBO significantly improved regeneration of cartilage in knee joint.
- „In the introduction, include more literature about existing treatments options for OA, then allogenic stem cell treatments, and hyperbaric oxygenation treatments, then challenges and advantages of allogenic stem cell treatments, and hyperbaric oxygenation treatments and line this up with the aim of current study.“
Response: Thank You for this question. As we have already mentioned in the Introduction section: „Currently available treatments for OA include weight control, exercise, and pharmacological approaches, which typically consist of intra-articularly injected viscoelastic supplements and analgesic therapies containing acetaminophen, salicylates, and nonsteroidal anti-inflammatory drugs.“
According to your suggestions, we have added another part related to OA therapy.
New introduction part: The standard pharmacological treatment includes agents for control of pain and inflammation (non-steroidal anti-inflammatory drugs, analgesics including opioids, intraarticular corticosteroids) and the group of the symptomatic slow acting drugs for OA such as glucosamine sulfate, chondroitin sulfate, diacerein, unsaponifiables extract of soybean and avocado administered orally and intrarticular hyaluronic acid (Hermann, W.; Lambova, S.; Muller-Ladner, U. Current Treatment Options for Osteoarthritis. Curr Rheumatol Rev. 2018, 14, 108-116.).
Cell therapy and tissue engineering have become increasingly common alternative treatments for cartilage defects (Lamoespinosa, J.M.; Mora, G.; Blanco, J.F.; et al. Intra-articular injection of two different doses of autologous bone marrow mesenchymal stem cells versus hyaluronic acid in the treatment of knee osteoarthritis: multicenter randomized controlled clinical trial (phase I/II). J Transl Med. 2016, 14, 246.).
Intra-articular injections may include administration of stem cells collected from different sources, platelet-rich plasma (PRP), hyaluronan preparations, and ozone (Joshi, J.N.; Rodríguez, L.; Reverté-Vinaixa, M.M.; Navarro, A. Platelet-Rich Plasma Injections for Advanced Knee Osteoarthritis: A Prospective, Randomized, Double-Blinded Clinical Trial. Orthop J Sports Med. 2017, 5, 2325967116689386).
Mesenchymal stem cells (MSC) are superior to others due to: the self-renewal ability; being essential for normal turnover and maintenance of cartilage; being capable to migrate to the damaged area of cartilage; and having the ability to induce chondrocyte proliferation and extracellular matrix (ECM) synthesis (Lamoespinosa, J.M.; Mora, G.; Blanco, J.F.; et al. Intra-articular injection of two different doses of autologous bone marrow mesenchymal stem cells versus hyaluronic acid in the treatment of knee osteoarthritis: multicenter randomized controlled clinical trial (phase I/II). J Transl Med. 2016, 14, 246.).
Until now, HBO therapy has been used for the treatment of numerous orthopedic diseases including soft tissue infections, acute traumatic ischemia, crushing, compartment syndrome, problematic wounds, refractory osteomyelitis, osteonecrosis, sports disorders, disease disorders and injuries (Lindell K Weaver (2014). Hyperbaric oxygen therapy indications, thirteenth edn. The Hyperbaric Oxygen Therapy Committee Report. Best Publishing Company, Florida). However, in the literature, there are limited number of preclinical studies which show the positive effects of HBO therapy on cartilage tissue (Yuan, LJ.; Ueng, S.W.; Lin, S.S.; Yeh, W.L.; Yang, C.Y.; Lin, PY. Attenuation of apoptosis and enhancement of proteoglycan synthesis in rabbit cartilage defects by hyperbaric oxygen treatment are related to the suppression of nitric oxide production. J Orthop Res. 2004, 22, 1126–1134. ;Ueng, S.W.; Yuan, L.J.; Lin, S.S.; Niu, C.C.; Chan, Y.S.; Wang, I.C. et al. Hyperbaric oxygen treatment prevents nitric oxide-induced apoptosis in articular cartilage injury via enhancement of the expression of heat shock protein. J Orthop Res. 2013, 3, 376–384. ; Nagatomo, F.; Gu, N.; Fujino, H.; Okiura, T.; Morimatsu, F.; Takeda, I.; Ishihara, A. Effects of exposure to hyperbaric oxygen on oxidative stress in rats with type II collagen-induced arthritis. Clin Exp Med. 2010, 1, 7–13).
However, Yılmaz et al. have shown that systemic medical O3 application was more effective than HBO therapy and may reduce development of cartilage damage and prevent OA formation (Yılmaz, O.; Bilge, A.; Erken, H.Y.; Kuru, T. The effects of systemic ozone application and hyperbaric oxygen therapy on knee osteoarthritis: an experimental study in rats. Int Orthop. 2021, 45, 2, 489-496.).
Recently published studies have shown that exposure of osteogenic-differentiating MSCs to HBO under in vitro simulated inflammatory conditions enhances differentiation towards the osteogenic phenotype, providing evidence of the potential application of HBO in all those processes requiring bone regeneration (Gardin, C.; Bosco, G.; Ferroni, L.; Quartesan, S.; Rizzato, A.; Tatullo, M.; Zavan, B. Hyperbaric Oxygen Therapy Improves the Osteogenic and Vasculogenic Properties of Mesenchymal Stem Cells in the Presence of Inflammation In Vitro. Int J Mol Sci. 2020, 20, 1452.).
A step further in the application of MSC and HBO as a co-treatment is shown by numerous clinical studies. In the therapy of certain diseases such as: multiple sclerosis, ALS, sports injuries and stroke, but there are no studies on the combined treatment with ADMSCs and HBO in OA knee joint regeneration (https://www.celixir.com/stem-cell-treatment-group-to-combine-treatment-with-hyperbaric-oxygen-therapy-hbot/).
In the methods: „Please include ethical approval statement, with ethical approval number, and the year approved.“
Response: Thank You for this suggestion. Please see the last paragraph of 2.3 subheadings in Material and methods section.
Results:
- „Figure 2 Legends: mention the panel A, and B. Same for Figure 3 and 4, i.e, the panel A, and B should be mentioned in the legends.“
Response: We completely agree with Your comment so we changed legends of Figures 2 to 7.
- „Figure 7D: How was the statistical difference between TN-alpha of day 21 and day 28. Similarly, how was the statistical difference between VEGFA quantified from day 21, and day 28. If it was non-significant mention the legends, or letter ns on the figures.“
Response: Thank You for this suggestion. We presented the results with bar charts but we changed them with dot plots, because is more informative then bar chart (please see new Figures 7). Also, statistically significant difference between the examined groups is already marked with ,,*”.
- „Figure 4 to figure 7: Why authors have not included the data from ADS alone, and untreated OA, an healthy controls. It is mandatory to compare the responses of treated groups with untreated OA, and healthy controls. Thus, I suggest authors to include the controls to draw the proper conclusions of the data.“
Response: Thank you for this very useful suggestion since undoubtedly addition of sham and positive groups would improve the quality of the paper and provide thorough information reffering to impact of OA on oxidative stress parameters as well as on inflammation. According to Your comment we included new group named as CTRL. Please see Figures 4, 5 and 6.
- „In the legends of all figures, with each panel, please indicate the number of replicates (n =?). and where statistical analysis is applied mention the name of tool or parameter applied.“
Response: Thank you for this very useful comment. Since the detailed information was not provided in Statistical analysis paragraph of our manuscript, this part remained confusing. The distribution of data was checked by Shapiro–Wilk test. In fact, we performed the one-way ANOVA and Bonferroni test for multiple comparisons in our research for comparison of parameters of systemic redox status which were determined in blood samples in one time point between four groups. Additionally, in our study data generated from time-course measurements such as changes in parameters of oxidative stress over the time (after 21 or 28days) were analyzed using two-way ANOVA followed by Bonferroni posttest to account for the two variables of time and treatment. Values of p < 0.05 were considered to be statistically significant. In order to clarify the tests used for statistical analysis we provided detailed and precise explanation in Material and Method section, please see paragraph Statistical analysis.
- „In figure panels showing bar graphs, especially Figure 2B, Figure 4 to 7. please replace the bar charts by dot plots to show the position/distribution of individual biological replicates.“
Response: Thank You for this suggestion. We change bar chart with dot plots, please see new Figures 4 to 7.
Others:
- „Please avoid control (CTRL) abbreviation in the abstract, if it was not used afterwards.“
Response: Thank You for this comment. We used this abbreviation throughout the text so we left it in the abstract section.
- „Similarly, in the last part of the abstract: either define the abbreviation IA, or better to write it intra-articular injection instead.“
Response: Thank You for this useful comment. We have integrated Your suggestion into abstract section. Please see page 2.
„Finally, the text style, founts, and grammar need to be corrected throughout the text.“
Response: Thank You for this comment. We standardized the font size, text style, founts, and grammar throughout the manuscript.

Round 2
Reviewer 1 Report
Authors did address most of the comments and I recommend it to accept it.
Reviewer 2 Report
The authors have made sufficient improvements to their manuscript, which is now suitable for publication.
I endorse the publication of this work. However, I have minor note for authors, which can be implemented during pdf proofs.
The word HBO was used as abbreviation in the first line of the abstract, but was defined in the later lines as hyperbaric oxygenation treatment (HBO).
Please define, when first time mentioned, and use abbreviation later on.